# Distinct Montages of Slow Oscillatory Transcranial Direct Current Stimulation (so-tDCS) Constitute Different Mechanisms during Quiet Wakefulness

**DOI:** 10.3390/brainsci9110324

**Published:** 2019-11-14

**Authors:** Ping Koo-Poeggel, Verena Böttger, Lisa Marshall

**Affiliations:** 1Institute of Experimental and Clinical Pharmacology and Toxicology, University of Lübeck, Ratzeburger Allee 160, 23562 Lübeck, Germany; chai.koo@uni-luebeck.de (P.K.-P.); verena.boettger@gmx.net (V.B.); 2Center for Brain, Behavior and Metabolism, University of Lübeck, Ratzeburger Allee 160, 23562 Lübeck, Germany

**Keywords:** slow oscillatory-tDCS, quiet wakefulness, stimulation parameters, tACS

## Abstract

Slow oscillatory- (so-) tDCS has been applied in many sleep studies aimed to modulate brain rhythms of slow wave sleep and memory consolidation. Yet, so-tDCS may also modify coupled oscillatory networks. Efficacy of weak electric brain stimulation is however variable and dependent upon the brain state at the time of stimulation (subject and/or task-related) as well as on stimulation parameters (e.g., electrode placement and applied current. Anodal so-tDCS was applied during wakefulness with eyes-closed to examine efficacy when deviating from the dominant brain rhythm. Additionally, montages of different electrodes size and applied current strength were used. During a period of quiet wakefulness bilateral frontolateral stimulation (F3, F4; return electrodes at ipsilateral mastoids) was applied to two groups: ‘Group small’ (*n* = 16, f:8; small electrodes: 0.50 cm^2^; maximal current per electrode pair: 0.26 mA) and ‘Group Large’ (*n* = 16, f:8; 35 cm^2^; 0.35 mA). Anodal so-tDCS (0.75 Hz) was applied in five blocks of 5 min epochs with 1 min stimulation-free epochs between the blocks. A finger sequence tapping task (FSTT) was used to induce comparable cortical activity across sessions and subject groups. So-tDCS resulted in a suppression of alpha power over the parietal cortex. Interestingly, in Group Small alpha suppression occurred over the standard band (8–12 Hz), whereas for Group Large power of individual alpha frequency was suppressed. Group Small also revealed a decrease in FSTT performance at retest after stimulation. It is essential to include concordant measures of behavioral and brain activity to help understand variability and poor reproducibility in oscillatory-tDCS studies.

## 1. Introduction

The application of weak electric current stimulation to the brain is a widely employed technique to induce functionally relevant changes in brain electric activity and/or behavioral or cognitive performance in human and animal models [1], and as such possess great therapeutic potential, e.g., [2,3,4]. However, application is hampered by variability and relatively poor reproducibility amongst studies, in particular for oscillating tACS or oscillatory- (o-) tDCS procedures [5]. While current strength and/or electrode size [6,7,8], and precise montage [9,10], have been reported to influence efficacy of tDCS, fewer studies have investigated modifications on tACS or so-tDCS [6,11,12,13,14]. 

The efficacy of weak electric stimulation on ongoing neural network activity can be assessed at the behavioral level or through physiological measures, most frequently EEG or MEP. While concordant effects of weak electric stimulation on EEG and behavior often occur, such as increased cortical excitability and increased performance [11,15,16,17,18,19], behavioral output is often measured alone [20,21,22], with stimulation-induced changes in brain electric network activity being implicitly assumed. 

Many previous studies, albeit not all [23,24,25], applying slow oscillation (so)-tDCS during sleep or during wakefulness with eyes-open, revealed concordant effects of stimulation on behavior and functionally associated EEG activity [3,18,26,27,28,29,30,31]. Here, we applied anodal so-tDCS during quiet wakefulness with eyes-closed to investigate whether so-tDCS would enhance frontal theta activity in a similar manner to the eyes-open state of both quiet and attentive wakefulness [27]. In addition to the most common montage used for so-tDCS in sleep and wakefulness studies [18,23,27,28,29,30,32,33], we employed a montage with larger electrodes and lower current density as often used in other contexts [34,35]. Moreover, to induce comparable vigilance states and cortical activity across sessions and subject groups, subjects performed a motor skill task [36].

## 2. Materials and Methods

### 2.1. Procedure

An adaptation session always preceded two experimental sessions. During the adaptation session, subjects arrived in the laboratory at 9 a.m. After EEG application subjects were to sit upright in a relaxed position with their eyes-closed in a well-lit room and to listen to exerts from two instrumental music pieces during a 30 min EEG recording period. After the session, the subject chose one of the pieces for both experimental sessions.

For each of the two experimental sessions, subjects also arrived in the laboratory at 9 a.m. After application of electrodes for EEG recording and so-tDCS subjects filled out the Standford Sleepiness Scale (SSS) and Positive and Negative Affect Schedule (PANAS) questionnaires and performed a finger sequence tapping task (see section: *Finger Sequence Tapping Task and Psychometric Questionnaires*). The total EEG recording period lasted ~60 min: Subsequent to a ~2 min settle-down and 15 min baseline period, a 30-min period followed in which either stimulation or sham-stimulation (subsequently only termed sham; see section: *So-tDCS*) was applied. Thereafter another 5 min of EEG were recorded. Subjects remained with their eyes-closed throughout the whole recording period and were instructed to mentally go through the task in their minds. EEG, EOG, and EMG were monitored continuously so that at any indication of drowsiness, a soft noise was introduced to maintain wakefulness. After the recording period, subjects were retested on the finger sequence tapping task and again filled out the SSS and PANAS questionnaires (Figure 1A).

After each experimental session, subjects were asked: “Did you feel anything on the scalp or elsewhere?’’, and “Do you think this was a stimulation session?”. The two conditions were pseudo-randomized and counterbalanced across subjects and sessions. The two sessions were at least 7 days apart. All participants signed a consent form prior to participation. The study was approved by the local ethics committee of the University of Luebeck, Germany.

### 2.2. Participants

Young healthy students aged between 18 and 28 years (mean age: 22.06 ± 0.41) from the university were recruited through flyers advertised around the campus. All participants were non-smokers, right-handed and native German-speakers. Presence or history of any form of sleep disturbance or cognitive impairment, any psychopathological disorder, seizures or brain injury, metal or cardiac pacemaker implantation, pregnancy and intake of medication except for contraceptive pills formed exclusion criteria. In total, 32 subjects participated; 16 (f:8) were assigned to ‘Group Small’ and 16 subjects (f:8) were assigned to ‘Group Large’ (Figure 1C,D; see section: *So-tDCS*). 

### 2.3. EEG Acquisition/Analysis

All signals were acquired with a DC amplifier SynAmps RT (Compumedics Neuroscan, Charlotte, NC, USA) at a sampling rate of 2 kHz, low pass filtered at 800 Hz, and amplified with a gain of 10 dB, resulting in an amplitude resolution of 32-bit float values, and accuracy of 29.80 nV/LSB. EEG was recorded using Ag/AgCl sintered electrodes from F7, Fz, F8, FCz, C3, Cz, C4, P3, Pz, P4, T5 and T6 (Figure 1B), referenced to the nose (International 10:20 system). To detect muscle and eye movements and to assess vigilance, submental electromyogram (EMG), vertical electrooculogram (EOG) from bipolar electrodes placed supraorbital and infraorbital, and horizontal EOG from the outer canthus of each eye were recorded. A ground electrode was positioned at Fpz. Impedance was kept below 5 kOhm for all electrodes.

The acquired EEG data were pre-processed using Brain Vision Analyzer (Gilching, Germany). Data were first low pass filtered using an infinite impulse response (IIR) at 30 Hz and high passed at 0.3 Hz (12 dB/oct), followed by down-sampling to 200 Hz based on spline interpolation. Independent component analysis (ICA) was applied to remove ocular artefacts [37] and corrected data were re-referenced to the average reference. Movement artefacts were excluded by visual inspection of 1 s epochs. Subsequent to artefact removal, data were extracted into eight 1 min long epochs: baseline 1 and baseline 2 (15–14 min and 1–2 min prior to onset of the stimulation period, t = 0 min, respectively), five stimulation free 1 min long periods (cp. section: *So-tDCS*), and one post-stimulation period (4–5 min after termination of the last stimulation). 

Power spectra were obtained for all recording locations using fast Fourier Transformation (FFT). Each 1 min epoch was segmented into 10 s windows, with 5 s overlap. Segments were zero-padded to the length of 10.24 s, with a Hanning window used for data tapering. The following frequency bands were analyzed: SO (0.7–1.2 Hz), Theta (4–8 Hz), and Alpha (8–12 Hz). Since individual alpha peak frequency (IAF) can vary strongly between subjects and has revealed greater sensitivity to task manipulations [38,39], analyses were also conducted for EEG power centered ± 1.25 Hz around the IAF. IAFs were estimated for anterior and centro-posterior locations separately as previously a different functional network between the frontal and parietal alpha band was suggested [40]. Frontal IAF was calculated as the average from peak alpha frequencies of the mean spectrograms at F7, Fz, FCz, and F8 within the range of 7–14 Hz from the total 15 min baseline period of the sham session (IAF = 9.29 ± 0.22 Hz, for subjects of both Groups). Similarly, IAF for calculations at central and parietal locations was obtained from peak alpha frequencies at P3, Pz, and P4 (IAF = 9.72 ± 0.18 Hz). For standard and IAF power, statistical comparisons between conditions were conducted after subtracting baseline level EEG power (mean of baselines 1 and 2), for each Group separately. 

As IAF can reveal state-dependent fluctuations across time [41] we compared IAF at baseline and during the post-stimulation epochs. There were no significant fluctuations of IAF across time, nor any interaction with the condition (Time main effect F (5, 75) < 2.01, *p* > 0.096; COND × TIME F (5, 75) < 1.13, *p* > 0.346). 

### 2.4. So-tDCS

The stimulation period consisted of anodal so-tDCS delivered bilaterally at dorsolateral frontal locations, F3, F4 of the international 10:20 system, with return electrodes placed at the corresponding ipsilateral mastoid in 5 epochs of 5 min followed by a stimulation free epoch of at least 1 min (Figure 1A,C,D). The first stimulation epoch commenced once a 15 min period of quiet wakefulness without any major movement artefacts occurred. In the sham session, at the end of a virtual 5 min stimulation block two subsequent slow oscillation cycles were delivered. The current was delivered by a customized battery-driven constant current stimulator with two synchronized circuits. Impedance was kept ≤ 1 kΩ.

For Group Small stimulator electrodes consisted of Ag/AgCl sintered cup electrodes of 8 mm diameter (Figure 1C), in Group Large electrodes were rectangular rubber stimulation electrodes of 7 × 5 cm (Figure 1D). The sinusoidal stimulation current (0.75 Hz) oscillated between 0 and a maximum value of 260 µA in Group Small. In Group Large a maximal current strength of 350 µA, below perception threshold was used. Calculated current densities under the electrodes were correspondingly 0.517 mA/cm² (Small) and 0.01 mA/cm² (Large). Total maximum current applied was therefore 0.52 mA for Group Small and 0.7 mA for Group Large. Stimulation electrodes were applied in an identical manner for both groups (i.e., disinfection and abrasion (EasyCap, Herrsching, Germany) of scalp locations, application of EC2 adhesive paste (Natus, Middleton, WI, USA) and use of additional tape for electrode fixation). 

### 2.5. Finger Sequence Tapping Task and Psychometric Questionnaires

The finger sequence tapping task (FSTT) was adapted from Walker et al. [42]. Subjects were instructed to use the fingers (except thumb) of their non-dominant hand to type a sequence of numbers (composition of five numbers ranging from 1–4) as quickly and accurately as possible. Training consisted of 12 blocks of learning each lasting 30 s and followed by a 30 s break interval. The retest, after the stimulation period, consisted of another three blocks of retrieval with the same sequence. The mean of the last 3 blocks of learning and the mean of the 3 blocks at retest was used for ANOVA analysis (see section: *Statistical Analysis)*.

Standford sleepiness scale (SSS) is a self-rating scale consists of eight progressive ordinal numbers where 1 corresponding to feeling active, vital, alert or wide awake and eight corresponding to asleep [43]. Subjects were to choose one item out of these orders that best described their sleepiness at the present moment when they filled out the questionnaire. The raw number that subjects rated was taken to analyze their sleepiness, thus the higher the number, the sleepier they are. Positive and negative affect schedule (PANAS) comprises ten positive and ten negative adjectives [44]. Subjects were to rate each item within the range of 1–5 (with 1 indicating very slightly or not at all, and 5 indicating extremely). Examples of adjectives are enthusiastic, or interested, for positive affect and distressed, or nervous for negative affect. Subjects were again instructed to rate these items based on the present moment when they filled out the questionnaire. The positive and negative items were then averaged separately to obtain single positive and negative values before and after the stimulation for analysis 

### 2.6. Statistical Analysis

For EEG power analysis (standard and IAF), repeated-measures, rmANOVAs using the factors: condition (COND) with levels SHAM and STIM; time (TIME) with the baseline-normalized five stimulation-free epochs and the post-stimulation epoch; and scalp topography (TOPO) with F7, Fz, F8, FCz, C3, Cz, C4, P3, Pz, P4, T5, T6 were used for each Group and frequency band separately. For post-hoc tests, comparisons between conditions were conducted using mean power of electrodes at frontal (F7, Fz, F8, Fcz), central (C3, Cz, C4), temporal (T5, T6) and parietal locations (P3, Pz, P4; Figure 1B). To directly compare the effect of the different stimulation parameters, we incorporated the factor: group (GROUP) for Groups Small and Large into the aforementioned ANOVA models. 

FSTT retention performance was analyzed using factors COND (SHAM, STIM) and TIME (training, retest) in a repeated-measures ANOVA. Differences over time in retention performance (retest-training) between sham and stim were assessed by post-hoc paired-sample t-tests without multiple comparison correction. 

All ANOVA results are Greenhouse-Geisser corrected for violation of the sphericity assumption (uncorrected degrees of freedom are given). All values are in mean ± SEM. A value *p* < 0.05 was considered significant.

### 2.7. Exploratory Theta Synchronization and Alpha Desynchronization

EEG theta synchronization (θ*_S_*) and alpha desynchronization (α*_DS_*) are reported to reveal a reciprocal relationship [45]. Since so-tDCS previously modified theta activity, we explored here the relationship between theta and alpha power after stimulation. In short, we used the formula:θ*_S_* = −(θ_Baseline_ − θ_PS_/θ_Baseline_) × 100where θ*_Baseline_* is standard baseline theta power before stimulation and θ*_PS_* is the mean theta power of the five stimulation-free epochs. Correspondingly, alpha desynchronization was calculated as: α*_DS_* = (α_Baseline_ − α_PS_/α_Baseline_) × 100

The sum θ*_S_* + α*_DS_* corresponds to the index of theta synchronization and alpha desynchronization. Thus, a greater positive value reflects both stronger θ*_S_* and α*_DS_*. ANOVA and post-hoc tests were conducted as described above. 

## 3. Results

### 3.1. Blinding of The Stimulation

In the stim session, four of the 32 subjects responded in the affirmative to the question ‘Do you think this was a stimulation session? (Three from Group Small), and two of them (one from Group Small) reported having felt something (i.e., tingling sensation on the head, tingling sensation under the eyes). In the sham session, 14 subjects reported affirmative to whether they believed to have undergone a stimulation session (six from Group Small). For sham, none of the subjects reported to have felt anything on their scalp, but six subjects (two from Group Small), reported other symptoms (e.g., muscle sore throughout the body, warm throat, uneasiness, tingling sensation on the hands). Hence, subjects were overall successfully blinded to the stimulation. 

### 3.2. Effect of Electrode Size and Current Strength on So-tDCS Induced Eeg Activity 

For Group Small, neither theta nor SO power revealed a significant main effect of COND, COND × TOPO or COND × TIME interaction in the rmANOVA (SO: COND main effect, F (1, 15) = 0.97, *p* = 0.340; COND × TOPO: F (11, 165) = 1.02, *p* = 0.374; COND × TIME: F (5, 75) = 1.26, *p* = 0.293; theta: COND main effect, F (1, 15) = 0.01, *p* = 0.923; COND × TOPO: F (11, 165) = 0.69, *p* = 0.573; COND × TIME: F (5, 75) = 1.70, *p* = 0.200). Alpha power, however, revealed a significant main effect of COND (F (1, 15) = 5.56, *p* = 0.032) and an interaction with topography (F (11, 165) = 4.68, *p* = 0.037; COND × TOPO), with post-hoc analysis indicating that so-tDCS decreased alpha power, in particular over the parietal region (Figure 2A). IAF power did not differ between conditions (F (1, 15) = 0.20, *p* = 0.664; COND × TOPO interaction (F (11, 165) = 0.30, *p* = 0.644; Figure 3A). 

For Group Large, analyses of theta power revealed a trend toward increased theta band power after so-tDCS compared to sham (COND main effect: F (1, 15) = 3.59, *p* = 0.078). In contrast to Group Small, IAF was modified by so-tDCS (COND (F (1, 15) = 6.26, *p* = 0.024), indicating that IAF power was decreased after so-tDCS as compared to sham, and limited to the parietal region (T (15) = 2.72, *p* = 0. 016; COND × TOPO: (F (11, 165) = 5.96, *p* = 0.014; Figure 2B). SO and standard alpha band power revealed neither a significant main effect of COND nor an interaction with time or topography (SO: COND main effect: F (1, 15) = 0.04, *p* = 0.849; COND × TOPO: F (11, 165) = 0.38, *p* = 0.748; COND × TIME: F (5, 75) = 1.35, *p* = 0.273; alpha: COND main effect: F (1, 15) = 0.17, *p* = 0.688; COND × TOPO: F (11, 165) = 0.34, *p* = 0.622; COND × TIME: F (5, 75) = 0.63, *p* = 0.606; see Appendix A for an overview of ANOVA results). 

### 3.3. So-tDCS Effect on Behavioral Parameters

As the FSTT was conducted to induce comparable vigilance states and cortical activity across sessions and subject groups, we had not expected any modification by so-tDCS to occur. However, for sham both groups showed an increase in performance with the passage of time (Time main effect for Group Large: F (1, 15) = 9.34, *p* = 0.008; Time main effect for Group Small: F (1, 15) = 8.74, *p* = 0.010).

Interestingly, performance was significantly modified by so-tDCS, however only in Group Small (COND × TIME: F (1, 15) = 5.69, *p* = 0.031; Figure 4A). To investigate whether the detrimental effect on behavior was specific to so-tDCS, in a supplementary experiment (*n* = 9), we applied IAF-tDCS using similar parameters as for Group Small. Performance was not decreased (Appendix A). In Group Large, stimulation did not affect performance (F (1, 15) = 0.24, *p* = 0.634; Figure 4B).

In Group Small subjects felt more positive after the anodal so-tDCS than sham session as revealed by PANAS (COND × TIME: F (1, 15) = 6.86, *p* = 0.019). Similarly, the SSS revealed that subjects felt more awake after anodal so-tDCS compared to sham (F (1, 15) = 10.08, *p* = 0.006), although a baseline difference might have contributed to this effect (sham: 2.38 ± 0.16; stim: 3.00 ± 0.26, T (15) = 2.61, *p* = 0.020).

Subjects in Group Large did not show any interaction for affect or sleepiness scales. In the so-tDCS session subjects marked slightly higher SSS scores (COND main effect: F = (1, 15) = 5.00, *p* = 0.041; stim 2.44 ± 0.26 vs. sham 2.06 ± 0.20), however, separate post-hoc tests, comparing scores between conditions, at the beginning and at the end of the experiment did not reach significance (*p* > 0.62 for each; see Appendix A). Also, post-hoc correlations of behavioral measures with both standard alpha and IAF power did not reveal any significant relationships (Pearson correlations, *p* > 0.1). 

### 3.4. Contrasting Effects of Stimulation Parameters

An ANOVA incorporating a GROUP factor for Group Small and Group Large supported the above results, that the suppression of standard alpha power after so-tDCS was more pronounced in Group Small than in Group Large (COND × GROUP: F (1, 30) = 4.83, *p* = 0.036, Figure 2A). None of the other frequency bands, including IAF, showed any indication of a group difference (*p* > 0.15). Moreover, independent sample t-tests did not reveal any significant difference in baseline power for any of the frequency bands. Training performance on the finger sequence tapping task did not differ between Groups (*p* > 0.57). 

### 3.5. Increased Theta Synchronization in Conjunction with Alpha Desynchronization 

For Group Large, a modulation by so-tDCS of EEG activity was underscored by the exploratory analyses of theta synchronization—alpha desynchronization index (COND × TOPO: F(11,165) = 2.78, *p* = 0.032) over the parietal region (stim: 28.25 ± 9.83, sham: 8.73 ± 7.02). The theta synchronization—alpha desynchronization index was not modified in Group Small (COND × TOPO: F(11, 165) = 0.55, *p* > 0.69). Also, post-hoc correlations of behavioral measures with the index did not reveal any significant relationships (Pearson correlations, *p* > 0.1). 

## 4. Discussion

In the present study, anodal so-tDCS applied bilaterally over the dorsolateral prefrontal cortex during quiet wakefulness with eyes-closed only tended to effect EEG theta power as was previously reported for wakefulness with eyes-open and failed to modulate ongoing SO power [27]. However, the suppression of the alpha power (i.e., alpha band or IAF) by so-DCS in both Groups, as well as the increase in the theta synchronization—alpha desynchronization index indicate, that the applied stimulation in the present study was indeed efficient. The modulatory effect of stimulation on parietal alpha power during wakefulness with eyes-closed can be interpreted as supporting brain state specific responsiveness to weak electric stimulation since the occipito-parietal alpha rhythm is dominating during the eyes-closed-state of wakefulness. Weak electric current stimulation has been found to modify, but not to change the frequency of, ongoing network activity [46,47,48,49]. The absence of significantly augmented theta and SO power in the present study as was found for wakefulness with eyes-open is most likely due to the pervasiveness of network mechanisms underlying the alpha rhythm in the eyes-closed state.

Of interest are the differences in EEG responses obtained for the two Groups. In Group Small, standard alpha band power relative to the baseline was suppressed by so-tDCS, Group Large showed a decrease in the narrow IAF power. Studies on standard alpha band activity have led to the concept that this cortical rhythm is the result of wide-spread cortical network activity mediated by cortico-thalamic loops [50,51,52]. A higher current density despite relatively small electrodes of Group Small may thus have been more efficient in modulating the activity of neurons belonging to functionally more diverse networks [53,54,55]. In contrast, the large electrodes with lower current density may have, more efficiently modulated the most dominant ongoing network activity, reflected by a narrow peak frequency. The pressing question is, what exactly these two alpha measurements reflect. Unfortunately, comparative investigations are rather limited. Standard alpha band and IAF are generally both linked to cognitive ability and/or capacity and are not typically reported together [38]. In investigating the genetic variation of the alpha rhythm, Smit and colleagues concluded that the genetic influences on IAF and alpha power were largely independent and these two bands may indicate distinct neural processes [56]. Speculating that endogenous IAF power reflects the activity of rather homogenous thalamo-cortical loops [57], our data tend to suggest that larger electrodes, at lower current density are more efficient at selectively modulating the predominantly engaged neural networks processing than a small electrode of higher current density. 

Nevertheless, our results are difficult to reconcile with those of D’Atri et al. who also applied a protocol of the anodal oscillating-tDCS at 0.8 Hz during eyes-closed resting state and found a qualitative increase in alpha power [58]. However, several parameters differed between the studies. The study by D’Atri et al. did not involve a foregoing learning period, which can affect responsiveness of the network [59,60]. Moreover, only two electrodes were employed in their study, one of which was an extracephalic reference. Although the present montage of stimulation electrodes was used as in a previous study in which frontal theta and slow oscillations during wakefulness were affected [27], the placement of the return electrodes over the mastoids, especially in Group Large, may have been relevant for effects on posterior alpha activity.

Although a change in performance on the motor skill task as a result of so-tDCS was not the main objective, performance at recall differed between groups, with an impairment found in Group Small, but not in Group Large. While it has been evidenced that tDCS applied over motor cortex during motor task could increase the performance, likely due to the facilitation of LTP [61], our stimulation applied bi-frontally had a detrimental effect. The stimulation may have interfered with the ongoing frontoparietal network; a network found to be increased after motor learning [62]. Indeed, the neural processes after learning in motor skill continue to evolve with the passage of time (see review [63,64]) and this process is susceptible to disruption [60]. Thus, we believe impaired motor skill at retest in Group Small was due to the interference of stimulation with the ongoing functional networks reflected in standard alpha band activity. The increased behavioral state of vigilance, pronounced interestingly also only in Group Small, may similarly reflect interference of the anodal oscillatory stimulation with the ongoing resting functional network [65,66]. Since the underlying generators of alpha rhythms as well as the dependence of the FSTT retention on brain rhythms are still under intense investigation [67,68,69], it would go beyond the scope of this paper to discuss the interaction between stimulation induced decrease in alpha and behavior. Interestingly, performance was maintained in Group Large, which may have benefitted from processes underlying the significant enhancement of theta synchronization plus alpha desynchronization found in this Group [45,70]. 

Finally, we should address a few limitations. Firstly, current density under each of the two electrode pairs was very low in Group Large, which may have contributed to the different effects. However, total applied current, a decisive measure for modelling putative effects of stimulation (e.g., [71]), was higher in Group Large than Group Small. Secondly, the lack of sleep schedule control prior to the experimental sessions restricted our interpretation on the baseline difference on sleepiness scale in Group Large. Moreover, these baseline differences potentially introduced additional inter- and intra-individual variation, which are known to affect the impact of tDCS (e.g., [72]). Lastly, the statistic used was not a conservative control for type I error, hence, the small effect of so-tDCS during eyes-closed wakefulness should be validated by further studies.

## 5. Conclusions

In summary, so-tDCS applied during wakefulness with eyes-closed modulated the dominant oscillation, thus, further underlining brain state specific responsiveness to so-tDCS and weak electric stimulation in general. Results also indicate a discordant response of EEG and behavior to weak electric stimulation applied over bilateral frontal cortex, for montages differing in electrodes size and in applied current strength. A differential impact on neural networks is presumed to underlie the variations in EEG and behavior. For future studies of weak electric current stimulation, we conclude, it is essential to include concordant measures of behavioral and brain activity as an important step toward understanding variability and poor reproducibility in the field.

## Figures and Tables

**Figure 1 brainsci-09-00324-f001:**
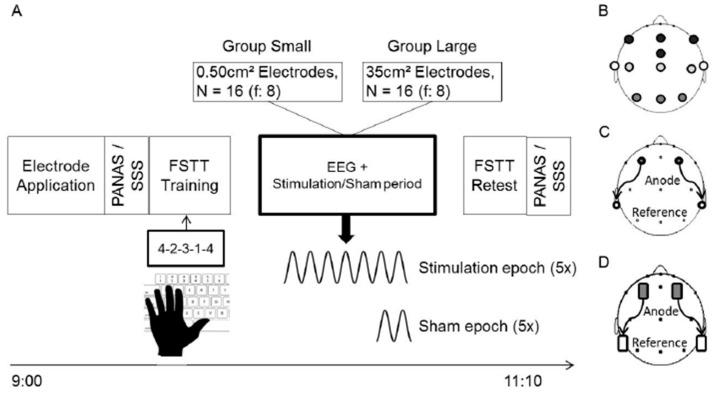
A schematic view of the experimental procedure. (**A**) Timeline of the experimental procedure for Group Small and Group Large. (**B**) Locations of EEG electrodes, and stimulation electrodes for Group Small (**C**) and Group Large (**D**). Open circles and rectangles in (**C**,**D**) represent the return electrodes. PANAS, Positive and Negative Affective Scale; SSS, Stanford Sleepiness Scale; FSTT, Finger sequence tapping task.

**Figure 2 brainsci-09-00324-f002:**
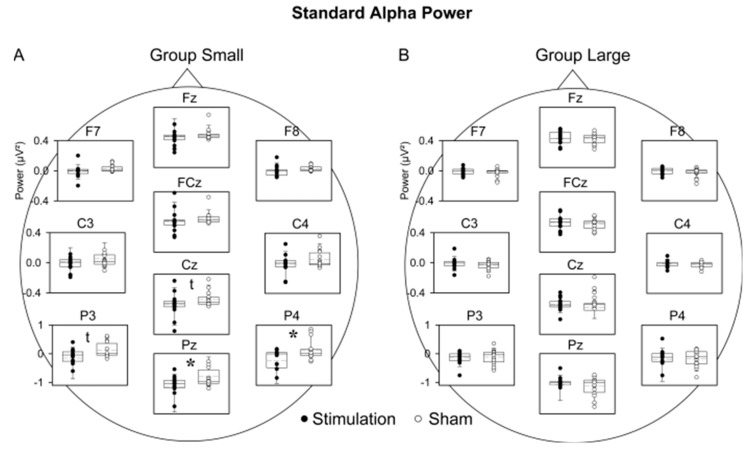
Comparison of standard alpha power between stimulation and sham conditions for Group Small and Group Large. (**A**) For Group Small, alpha power differed significantly at Pz and P4 and showed a tendency to differ at Cz and P3. When data for all three parietal electrodes were pooled alpha power in the stimulation condition was likewise decreased in sham (*p* = 0.029). (**B**) For Group Large there were no significant differences between conditions. For clarity temporal sites (T5, T6) are not shown. Each box plot shows the median as the horizontal line, with the bottom and top whiskers representing 10th and 90th percentiles, respectively. Circles represent the individual subject. * *p* < 0.05; t *p* < 0.10.

**Figure 3 brainsci-09-00324-f003:**
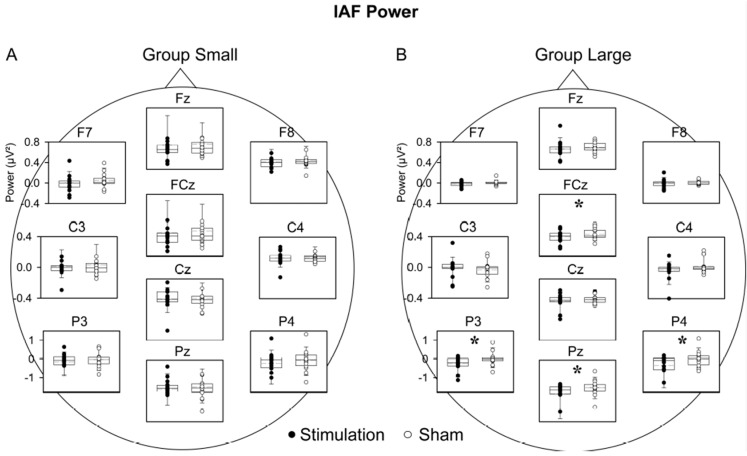
Comparison of IAF power between stimulation and sham conditions for Group Small and Group Large. (**A**) For Group Small there were no significant differences between conditions. (**B**) Group Large revealed significantly lower IAF power in the stimulation than sham condition at FCz and the parietal region. For clarity temporal sites (T5, T6) are not shown. Each box plot shows the median as the horizontal line, with the bottom and top whiskers representing 10th and 90th percentiles, respectively. Circles represent the individual subject. * *p* < 0.05.

**Figure 4 brainsci-09-00324-f004:**
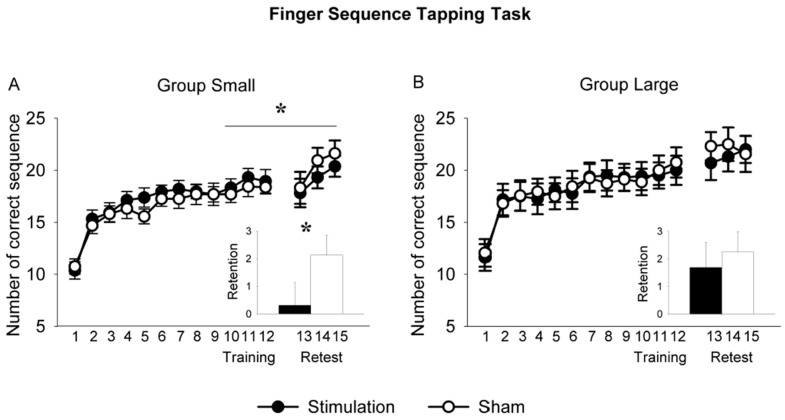
Training and retest performance on the finger sequence tapping task in the conditions stimulation and sham for Group Small (**A**) and Group Large (**B**). The x-axis represents the number of correct sequences for Training (trials 1–12) and Retest (trials 13–15). In the insets “Training” refers to the average of sequences 10–12 and “Retest” to the average of sequences 13–15. In Group Small improvement on the task was significantly reduced in the stimulation condition compared to sham. This reduction was not observed in Group Large. Asterisk and vertical line represent the significant COND × TIME interaction; the asterisk in the bar chart depicts the significant difference of the paired-sample t-test. * *p* < 0.05.

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
