# Peer review of "Distinct Montages of Slow Oscillatory Transcranial Direct Current Stimulation (so-tDCS) Constitute Different Mechanisms during Quiet Wakefulness"

_brainsci, 2019, doi:10.3390/brainsci9110324_

Round 1

Reviewer 1 Report

The study examined the effect on EEG activity of two anodal so-tDCS montages with different electrodes size and applied current strength. The stimulation has been applied in eyes-closed condition bilaterally on dorso-lateral frontal areas (F3,F4) with the return electrodes placed on the ipsilateral mastoids. The montages have been tested on two different groups and compared with a sham condition. The main result includes a modulatory effect of so-tDCS on parietal alpha power, with the small-electrodes montage (with higher current density) reducing the spectral power in the whole frequency band, while the stimulation vialarger electrodes reduced the power only at the Individual Alpha Peak frequency.

The topic of the study, i.e. the investigation on stimulation parameters influencing the responsiveness to NIBS, is of interest for the scientific community of the field. The experimental design is well controlled and the involvement of objective, subjective and behavioral measures to asses the stimulation effects is a merit.

Indeed, this methodology allowed the authors to describe also a behavioral detrimental effect of so-tDCS on the finger tapping task used to induce comparable cortical activity across sessions and subject groups, that is present in the case of stimulation with small electrodes only. In the same group, so-tDCS also induced more positive feelings (as revealed by PANAS) and reduced sleepiness (as assessed by Stanford Sleepiness Scale) suggesting a possible relationship between these effects and those on the EEG alpha power.

In my opinion, the work is suitable for the publication after minor revisions.

Comments:

Materials and Methods

Line 60. Are the active and sham stimulation performed in different days? It should be detailed in the section 2.1 Procedure, along with the time elapsing between the two sessions.

Line 141 The maximal current strength for the Larger electrodes montage at 350 microA , corresponding to a current density of 0.01 mA/cm2, seems to be too low as compared to that of the small electrodes. Accordingly, the stimulation could have stronger EEG and behavioral effect on the Group Small than on the Large one due to usage of a too “weak” stimulation in the latter group. Indeed, in many studies adopting 7 x 5 cm electrodes, the applied current density is of 1/1.5 mA. In my opinion, a similar current intensity would be able to induce comparable (or stronger) effects than those shown in the Group Small.

Line 84. Did the authors control for the participants sleep schedule (and/or quality) of the night preceding the two experimental days? Results on subjective sleepiness reported a significant difference at the baseline between sham and so-tDCS condition for the Group Small (that also showed a significant TIME x COND interaction) and a significant main effect of the condition for the Group Large (with no difference at baseline nor post-stimulation).

Differences between conditions and groups in the sleep pressure could be considered a strong source of variability in the assessment of subjective, behavioral and EEG effects of the so-tDCS  and should be discussed.

Line 148. It could be useful for the readers to have a brief description also for PANAS and SSS, as it has been done for FSTT.

Line 156 2.6 Statistical analyses. A multiple-comparisons correction should be applied on the ANOVAs comparing the EEG data.

Line 161. The authors should explicit how the post-hoc comparisons on topography has been performed. Were data averaged among the different scalp locations belonging to same area and then compared across areas?

Results

Line 228 Is there a correlation between the effects on EEG (alpha power/thetaS + alphaDS) and those on behavioral task and self-reported measures in the Group Small? In my experience, in an eyes-closed condition the gradual disappearing of alpha oscillations is a sign of increasing sleepiness and of approaching to sleep onset. In this view, the decrease in alpha power induced by so-tDCS could be related to the worsening of the performance at the behavioral task due to the induction of increased sleep pressure by the modulation of the top-down control system of sleep-wake cycle. However, it is hard to reconcile this view with the significant reduction in sleepiness level reported by the subjects. Could the author discuss this issue?

Is there any correlation between the changes in IAF and changes in the performance, PANAS and SSS scores? please, evaluate it in both the groups and discuss on possible differences.

Discussion

A section with the limits of the study should be included in the discussion.

Author Response

Dear Reviewer,

Thank you very much for your productive comments. We believe we have addressed all points accordingly (see below).

Sincerely,

Ping Koo-Poeggel

Comments and Suggestions for Authors

The study examined the effect on EEG activity of two anodal so-tDCS montages with different electrodes size and applied current strength. The stimulation has been applied in eyes-closed condition bilaterally on dorso-lateral frontal areas (F3,F4) with the return electrodes placed on the ipsilateral mastoids. The montages have been tested on two different groups and compared with a sham condition. The main result includes a modulatory effect of so-tDCS on parietal alpha power, with the small-electrodes montage (with higher current density) reducing the spectral power in the whole frequency band, while the stimulation vialarger electrodes reduced the power only at the Individual Alpha Peak frequency.

The topic of the study, i.e. the investigation on stimulation parameters influencing the responsiveness to NIBS, is of interest for the scientific community of the field. The experimental design is well controlled and the involvement of objective, subjective and behavioral measures to asses the stimulation effects is a merit.

Indeed, this methodology allowed the authors to describe also a behavioral detrimental effect of so-tDCS on the finger tapping task used to induce comparable cortical activity across sessions and subject groups, that is present in the case of stimulation with small electrodes only. In the same group, so-tDCS also induced more positive feelings (as revealed by PANAS) and reduced sleepiness (as assessed by Stanford Sleepiness Scale) suggesting a possible relationship between these effects and those on the EEG alpha power.

In my opinion, the work is suitable for the publication after minor revisions.

Reviewers’ Comments:

Materials and Methods

Line 60. Are the active and sham stimulation performed in different days? It should be detailed in the section 2.1 Procedure, along with the time elapsing between the two sessions.

Authors Response:

Thank you for your comment. We have inserted a new sentence on l. 80. The text now reads:

“ … counterbalanced across subjects and sessions. The two sessions were at least 7 days apart”

 Reviewers’ Comments:

Line 141 The maximal current strength for the Larger electrodes montage at 350 microA , corresponding to a current density of 0.01 mA/cm2, seems to be too low as compared to that of the small electrodes. Accordingly, the stimulation could have stronger EEG and behavioral effect on the Group Small than on the Large one due to usage of a too “weak” stimulation in the latter group. Indeed, in many studies adopting 7 x 5 cm electrodes, the applied current density is of 1/1.5 mA. In my opinion, a similar current intensity would be able to induce comparable (or stronger) effects than those shown in the Group Small.

Authors Response:

Yes, we agree that the current density under each electrode of the large electrodes was very low, and that a higher value would have enabled a more straightforward comparison to Group Small. Moreover, we are aware, that only about half the amount of applied current is said to enter the brain. However, next to current density the total amount of injected current is decisive for modelling putative effects of stimulation (e.g. Datta et al 2010). For Group Large 0.7 mA were applied (total current of both electrode pairs), whereas for Group Small this was only 0.52 mA. Since when piloting, two subjects reported feeling uncomfortable (not cutaneous) when a maximum of 400 µA was, we reduced the maximum current strength in Group Large.

We have introduced this information into an additional paragraph on limitations, as requested, l. …:

 “… limitations. Firstly, current density under each of the two electrode pairs was very low in Group Large, which may have contributed to the different effects. However, total applied current, a decisive measure for modelling putative effects of stimulation (e.g. Datta et al 2010), was higher in Group Large than Group Small. Secondly, …”

Reviewers’ Comments:

Line 84. Did the authors control for the participants sleep schedule (and/or quality) of the night preceding the two experimental days? Results on subjective sleepiness reported a significant difference at the baseline between sham and so-tDCS condition for the Group Small (that also showed a significant TIME x COND interaction) and a significant main effect of the condition for the Group Large (with no difference at baseline nor post-stimulation).

Differences between conditions and groups in the sleep pressure could be considered a strong source of variability in the assessment of subjective, behavioral and EEG effects of the so-tDCS and should be discussed. (Datta et al., 2010)

Authors’ Response:

Thank you for raising this concern. We assessed per questionnaire typical sleep/wake schedule, but unfortunately did not control for the sleep schedule of the night preceding the two experimental days. A regular sleep /wake schedule was a prerequisite for participation.  We agree that the significant subjective difference in sleepiness may potentially have influenced results. We address this issue in the paragraph on study limitations:

 “… effects. Secondly, the lack of sleep schedule control prior to the experimental sessions restricted our interpretation on the baseline difference on sleepiness scale in Group Large. Moreover, these baseline differences potentially introduced additional inter- and intra-individual variation, which are known to effect the impact of tDCS (e.g. Koo et la 2018). Lastly,…”

 Reviewers’ Comments:

Line 148. It could be useful for the readers to have a brief description also for PANAS and SSS, as it has been done for FSTT.

Authors’ Response:

Thank you. We have added a paragraph to section to 2.5. , which is now subtitled 2.5. Finger sequence taping task and psychometric questionnaires” to describe the scales used and their content at line xx and added corresponding citations (l. 156):

“Standford sleepiness scale is a self-rating scale consists of 8 progressive ordinal numbers where 1 corresponding to feeling active, vital, alert or wide awake and 8 corresponding to asleep (Hoddes et al., 1973). Subjects were to choose one item out of these orders that best described their sleepiness at the present moment when they filled out the questionnaire. The raw number that subjects rated was taken to analyze their sleepiness, thus the higher the number, the more sleepy. Positive and negative affection scale (PANAS) comprises 10 positive and 10 negative adjectives (Watson et al., 1988). Subjects were to rate each item within the range of 1 to 5 (with 1 indicating very slightly or not at all, and 5 indicating extremely). Examples of adjectives are enthusiastic, or interested, for positive affection and distressed, or nervous for negative affection. Subjects were again instructed to rate these items based on the present moment when they filled out the questionnaire. The positive and negative items were then averaged separately to obtain single positive and negative values before and after the stimulation for analysis.”

Reviewers’ Comments:

Line 156 2.6 Statistical analyses. A multiple-comparisons correction should be applied on the ANOVAs comparing the EEG data.

Authors’ Response:

Although we agree this is a logical consideration, as far as we know it is not customary to apply multiple-comparisons correction to these ANOVAS, since for each frequency band a separate hypothesis is implicitly applied. That fact that our reported effects are weak is expressed in the limitations paragraph.

Reviewers’ Comments:

Line 161. The authors should explicit how the post-hoc comparisons on topography has been performed. Were data averaged among the different scalp locations belonging to same area and then compared across areas?

Authors’ Response:

In order to avoid multiple post-hoc testing we performed post-hoc analyses on effects between conditions on pooled data of presumed functionally cohesive regions (mean of frontal, central, parietal and temporal electrodes, respectively). A sentence has been modified to clarify the averaged of the corresponding electrodes under Statistical analysis at line 172.

“For post-hoc tests, comparisons between conditions were conducted using mean power of electrodes at frontal (F7, Fz, F8, Fcz), central (C3, Cz, C4), temporal (T5, T6) and parietal locations (P3, Pz, P4; Figure 1B).”

Reviewers’ Comments:

Results

Line 228 Is there a correlation between the effects on EEG (alpha power/thetaS + alphaDS) and those on behavioral task and self-reported measures in the Group Small? In my experience, in an eyes-closed condition the gradual disappearing of alpha oscillations is a sign of increasing sleepiness and of approaching to sleep onset. In this view, the decrease in alpha power induced by so-tDCS could be related to the worsening of the performance at the behavioral task due to the induction of increased sleep pressure by the modulation of the top-down control system of sleep-wake cycle. However, it is hard to reconcile this view with the significant reduction in sleepiness level reported by the subjects. Could the author discuss this issue?

Is there any correlation between the changes in IAF and changes in the performance, PANAS and SSS scores? please, evaluate it in both the groups and discuss on possible differences.

 Authors’ Response:

On correlations:

We now correlated (i) alpha desynchronization and theta synchronization with the behavioral task as well as the self-reported questionnaires, and (ii) changes in IAF and changes in the performance, PANAS and SSS scores. None of the analyses yielded significant correlations between these parameters. Since a discussion on this would be too speculative, we omit this in the discussion. Also, in the case of (i) we did not expect modifications to be direct enough to yield a correlation, and have previously rarely found correlations with behavioral control parameters.

Results of the correlations are given in the results section “So-tDCS effect on behavioural parameters”, line 270

“… (see Supplementary Note 2). Also, post-hoc correlations of behavioral measures with standard alpha and IAF did not reveal any significant relationships (Pearson correlations, p > 0.1).”

And line 285:

“… p > 0.69). Also, post-hoc correlations of behavioral measures with the index did not reveal any significant relationships (Pearson correlations, p > 0.1). “

On alpha and task performance:

Thank you for raising this point. We also noticed this discordance, but had avoided speculating on it as the role of post-learning alpha on the recall performance FSTT is not clear (Humiston & Wamsley 2018, Bönstrup et al 2019). While it is true that the disappearing of alpha oscillations could imply a progressive drowsiness that will lead to sleep onset, it was however also suggested that different alpha rhythms as well as generators can represent distinct neural processes (e.g, Ben-Simon et al, 2008).

We now address the issue in the underlined paragraph (l. 319):

 “…ongoing resting functional network [62, 63].

Since the underlying generators of alpha rhythms as well as the dependence of the FSTT retention on brain rhythms are still under intense investigation (Ben-Simon et al., 2008; Humiston & Wamsley, 2018; Bönstrup et al., 2019), it would go beyond the scope of this paper to discuss the interaction between stimulation induced decrease in alpha and behavior. Interestingly, performance…”

Reviewers’ Comments:

Discussion

 A section with the limits of the study should be included in the discussion.

Authors’ Response:

We have added Limitations into a new paragraph before the Conclusion:

 “Finally, we should address a few limitations. Firstly, current density under each of the two electrode pairs was very low in Group Large, which may have contributed to the different effects. However, total applied current, a decisive measure for modelling putative effects of stimulation (e.g. Datta et al 2010), was higher in Group Large than Group Small. Secondly, the lack of sleep schedule control prior to the experimental sessions restricted our interpretation on the baseline difference on sleepiness scale in Group Large. Moreover, these baseline differences potentially introduced additional inter- and intra-individual variation, which are known to effect the impact of tDCS (e.g. Koo et al 2018). Lastly, the statistic used was not a conservative control for type I error, hence, the small effect of so-tDCS during eyes-closed wakefulness should be validated by further studies.”

Ben-Simon, E., Podlipsky, I., Arieli, A., Zhdanov, A. & Hendler, T. (2008) Never resting brain: simultaneous representation of two alpha related processes in humans. PLoS One, 3, e3984.

Bönstrup, M., Iturrate, I., Thompson, R., Cruciani, G., Censor, N. & Cohen, L.G. (2019) A Rapid Form of Offline Consolidation in Skill Learning. Curr Biol, 29, 1346-1351 e1344.

Datta, A., Bikson, M. & Fregni, F. (2010) Transcranial direct current stimulation in patients with skull defects and skull plates: high-resolution computational FEM study of factors altering cortical current flow. Neuroimage, 52, 1268-1278.

Hoddes, E., Zarcone, V., Smythe, H., Phillips, R. & Dement, W.C. (1973) Quantification of sleepiness: a new approach. Psychophysiology, 10, 431-436.

Humiston, G.B. & Wamsley, E.J. (2018) A brief period of eyes-closed rest enhances motor skill consolidation. Neurobiology of learning and memory, 155, 1-6.

Watson, D., Clark, L.A. & Tellegen, A. (1988) Development and validation of brief measures of positive and negative affect: the PANAS scales. Journal of personality and social psychology, 54, 1063.

Reviewer 2 Report

This paper from Koo-Poeggel and colleagues describes how exposure to slow oscillatory tDCS (so-tDCS) influences brain electrical activity and behavioral measures.

The manuscript is of interest and scientifically sound even if the effects of so-tDCS are still controversial.

There are some limitations that could be adressed and the paper could be improved to reach the interest of more readers.

In the text there some typos and punctuation errors that should be revised A table including results from statistical analysis could help making them more clear The part of the discussion about alpha band activity modification is too speculative – the effect of such small amplitude currents is not thought to modify the activity of large neural network involving cortical-subcortical structures. I suggest to be more prudent when discussing the effect of small/large electrodes (even considering the spatial dispersion of the electric current). Besides, limitations of the study should be more clearly stated in the discussion (e.g. the effect of inter- and intra-individual variability on tDCS).

In my overall opinion, this paper could be accepted for a publication with some revisions.

Author Response

Dear Reviewer,

Thank you very much for your insightful comments.

We believe we have addressed all points accordingly (see below).

Sincerely,

Ping Koo-Poeggel

Comments and Suggestions for Authors

This paper from Koo-Poeggel and colleagues describes how exposure to slow oscillatory tDCS (so-tDCS) influences brain electrical activity and behavioral measures.

The manuscript is of interest and scientifically sound even if the effects of so-tDCS are still controversial.

There are some limitations that could be adressed and the paper could be improved to reach the interest of more readers.

Reviewers’ Comments:

In the text there some typos and punctuation errors that should be revised.  

 Authors’ Response:

We have thoroughly read the text, and improved typos and punctuation.

 Reviewers’ Comments:

A table including results from statistical analysis could help making them more clear.

Authors’ Response:

Thank you. We have added two tables with most relevant results of the ANOVAs for EEG and behavior. However, as it duplicates the information given in the main text, we put it into Supplementary information, as Note 3.

Table 1

Test

Effect

Group Small

Group Large

F value

P value

F value

P value

SO

Cond

0.97

0.34

0.04

0.85

Time

1.24

0.30

0.67

0.53

Cond x Topo

1.02

0.37

0.38

0.75

Cond x Time

1.26

0.29

1.35

0.27

Theta

Cond

0.01

0.92

3.59

0.08

Time

0.66

0.54

0.67

0.59

Cond x Topo

0.69

0.54

1.29

0.28

Cond x Time

1.70

0.20

0.31

0.84

Alpha

Cond

5.56

0.03*

0.17

0.69

Time

2.71

0.07

2.31

0.09

Cond x Topo

4.68

0.04*

0.34

0.62

Cond x Time

0.55

0.61

0.63

0.61

IAF

Cond

0.20

0.66

6.26

0.02*

Time

2.59

0.08

2.86

0.05

Cond x Topo

0.30

0.64

5.96

0.01*

Cond x Time

0.69

0.52

0.56

0.63

θS αDS

Cond

0.02

0.90

0.23

0.64

Cond x Topo

0.55

0.71

2.78

0.03*

Table 1. Overview of F statistics for relevant Condition and interactions with Condition of EEG data. All values were rounded up to two decimal points. Asterisks represent p < 0.05; Greenhouse-Geisser corrections were applied if necessary. Degrees of freedom, Cond: F(1,15); Time: F(5,75); Cond x Topo interaction: F(11,165), Cond x Time interaction: F(5, 75).

Table 2

Test

Effect

Small

Group Large

F value

P value

F value

P value

Positive

Cond

0.01

0.94

0.37

0.56

Cond x Time

6.86

0.02*

0.14

0.71

Negative

Cond

3.92

0.07

2.91

0.11

Cond x Time

2.76

0.12

0.26

0.62

SSS

Cond

0.63

0.44

5.00

0.04*

Cond x Time

10.08

0.006*

0.65

0.43

FSTT

Cond

0.06

0.81

0.50

0.49

Cond x Time

5.69

0.03*

0.24

0.63

Table 2. Overview of F statistics for relevant Condition and interactions with Condition of behavioral and control data. All values were rounded up to two decimal points. Asterisks represent p < 0.05; Greenhouse-Geisser corrections were applied if necessary. Degress of freedom, Cond: F(1,15);  Cond x Time interaction: F(1,15).

Reviewers’ Comments:

The part of the discussion about alpha band activity modification is too speculative – the effect of such small amplitude currents is not thought to modify the activity of large neural network involving cortical-subcortical structures. I suggest to be more prudent when discussing the effect of small/large electrodes (even considering the spatial dispersion of the electric current).

Authors’ Response:

Thank you for your suggestion. We have used rephrased the corresponding section referring now only more broadly to effects of stimulation on neural networks. We also added a reference by Schreckenberger et al. (2004) indicating thalamo-cortical activity in the generation of the alpha rhythm (line 313).

“Speculating that endogenous IAF power reflects the activity of rather homogenous thalamo-cortical loops (Schreckenberger et al 2004), our data tend to suggest that larger electrodes, at lower current density were more efficient at selectively modulating the predominantly engaged neural networks processing than a small electrode of higher current density.”

Furthermore, we address the weak effect of our applied stimulation in a new paragraph on study limitations. Please see Authors’ response to the comment below.  

Reviewers’ Comments:

Besides, limitations of the study should be more clearly stated in the discussion (e.g. the effect of inter- and intra-individual variability on tDCS).

 Authors’ Response:

We have now added a new paragraph on to describe potential caveats of this study (line 344).  

“Finally, we should address a few limitations. Firstly, current density under each of the two electrode pairs was very low in Group Large, which may have contributed to the different effects. However, total applied current, a decisive measure for modelling putative effects of stimulation (e.g. Datta et al 2010), was higher in Group Large than Group Small. Secondly, the lack of sleep schedule control prior to the experimental sessions restricted our interpretation on the baseline difference on sleepiness scale in Group Large. Moreover, these baseline differences potentially introduced additional inter- and intra-individual variation, which are known to affect the impact of tDCS (e.g. Koo et al. 2018). Lastly, the statistic used was not a conservative control for type I error, hence, the small effect of so-tDCS during eyes-closed wakefulness should be validated by further studies.”

 Reviewers’ Comments:

In my overall opinion, this paper could be accepted for a publication with some revisions.

Schreckenberger, M., Lange-Asschenfeld, C., Lochmann, M., Mann, K., Siessmeier, T., Buchholz, H.G., Bartenstein, P. & Grunder, G. (2004) The thalamus as the generator and modulator of EEG alpha rhythm: a combined PET/EEG study with lorazepam challenge in humans. Neuroimage, 22, 637-644.